# High Cell Selectivity and Bactericidal Mechanism of Symmetric Peptides Centered on d-Pro–Gly Pairs

**DOI:** 10.3390/ijms21031140

**Published:** 2020-02-08

**Authors:** Boyan Jia, Yiming Wang, Ying Zhang, Zi Wang, Xue Wang, Inam Muhammad, Lingcong Kong, Zhihua Pei, Hongxia Ma, Xiuyun Jiang

**Affiliations:** 1College of Animal Science and Technology, Jilin Agricultural University, Xincheng Street No.#2888, Changchun 130118, China; jiaboyan@jlau.edu.cn (B.J.); wym27149@jlau.edu.cn (Y.W.); wangxue@jlau.edu.cn (X.W.); dr.inam@sbbu.edu.pk (I.M.); lingcong@jlau.edu.cn (L.K.); peizhihua@jlau.edu.cn (Z.P.); 2College of Animal Science and Technology, Inner Mongolia University for Nationalities, Tongliao 028000, China; wangzi0225@jiau.edu.cn; 3College of Life Science, Jilin Agricultural University, Xincheng Street No.#2888, Changchun 130118, China; 4College of Animal Science and Technology, Changchun Sci-Tech University, Changchun 130600, China

**Keywords:** antimicrobial peptides, cell selectivity, stability, mechanism, endotoxin neutralization

## Abstract

Antimicrobial peptides (AMPs) have a unique action mechanism that can help to solve global problems in antibiotic resistance. However, their low therapeutic index and poor stability seriously hamper their development as therapeutic agents. In order to overcome these problems, we designed peptides based on the sequence template XXRXXRRzzRRXXRXX-NH_2_, where X represents a hydrophobic amino acid like Phe (F), Ile (I), and Leu (L), while zz represents Gly–Gly (GG) or d-Pro–Gly (pG). Showing effective antimicrobial activity against Gram-negative bacteria and low toxicity, designed peptides had a tendency to form an α-helical structure in membrane-mimetic environments. Among them, peptide LR_pG_ (X: L, zz: pG) showed the highest geometric mean average treatment index (GM_TI_ = 73.1), better salt, temperature and pH stability, and an additive effect with conventional antibiotics. Peptide LR_pG_ played the role of anti-Gram-negative bacteria through destroying the cell membrane. In addition, peptide LR_pG_ also exhibited an anti-inflammatory activity by effectively neutralizing endotoxin. Briefly, peptide LR_pG_ has the potential to serve as a therapeutic agent to reduce antibiotic resistance owing to its high therapeutic index and great stability.

## 1. Introduction

With the widespread use of antibiotics in clinical practice in recent years, the resistance rate of bacteria to traditional antibiotics is increasing. The development of antibacterial reagents with new antibacterial mechanisms is imminent [1]. As an important part of innate immunity, antimicrobial peptides (AMPs) form the first line of defense against antimicrobial infections, as found in many microorganisms including insects, plants, and animals [2]. Furthermore, AMPs are known for their anti-inflammatory properties, anti-biofilm formation, promotion of tissue and injury repair, and anti-cancer effects [2,3,4]. Compared to traditional antibiotics, AMPs exert an antibacterial effect through non-receptor-mediated membrane permeability. AMPs are expected to be alternatives to antibiotics because of their unique action mechanism [5,6,7]. Although more than 3000 AMPs were found, their application as therapeutic agents is seriously hindered due to some limiting factors including manufacturing costs, cytotoxicity, and poor stability [8].

The heptad repeat sequence is a repeat sequence composed of seven amino acids, where the “a” and “d” positions are occupied by hydrophobic amino acids [9]. It was found that natural AMPs possessing heptad repeat sequences have great antibacterial and anti-inflammatory activity, but their application is hindered by high cytotoxicity [10,11]. Previous studies mostly focused on how different hydrophobic amino acids at the “a” and “d” positions influence antibacterial activity and cytotoxicity of AMPs, but few studies used this sequence element to design new AMPs [12,13]. Some papers showed that symmetric AMPs centered on Gly–Gly (GG) or d-Pro–Gly (pG) have good antibacterial activity and high cell selectivity [14,15,16,17]. Therefore, we wondered if the same effect could be achieved when GG or pG was introduced into two symmetric heptad repeat sequences.

In this study, two symmetric heptad repeat sequences were connected by a short loop, and the sequence was designed as XXRXXRRzzRRXXRXX-NH_2_, where X represents Phe (F), Ile (I), and Leu (L), while zz represents Gly–Gly (GG) or d-Pro–Gly (pG). The net charge was +6 and the hydrophobicity was 40%–60%, which is consistent with the statistical information on natural AMPs [2,18]. Hydrophobic amino acids F, I, and L were used to damage microbial cell membranes [19]. Flexible (GG) and rigid (pG) loops were selectively inserted into the middle of two symmetric heptad repeat sequences to enhance cell selectivity [17,20]. In addition, Arg (R) was selectively introduced into the sequence template. The guanidine side chain of R allows it to form strong bidentate H-bonds with the phosphate moiety of lipid head groups, thereby enhancing the electrostatic action between peptides and anionic bacterial membranes [20]. To determine the role of loops in the designed peptides, a control peptide was designed as a heptad repeat sequence as follows: the central pG loop of peptide LR_pG_ with highest cell selectivity was removed, and Leu located at the N-terminus and C-terminus of the peptide was exchanged with Gly and d-Pro, respectively. The positive charge and hydrophobicity were consistent with LR_pG_. Finally, by amidating the C-terminus of the peptides, the charge of the AMPs was increased, and the stability of the structure was improved [21,22,23]. Using this method of designing and optimizing AMPs, we obtained LR_pG_ (X: L, zz: pG) with high cell selectivity and good conditional stability. The antibacterial mechanism study demonstrated that LR_pG_ played the role of an anti-Gram-negative bacterial agent through destroying the cell membrane.

## 2. Results

### 2.1. Peptide Design and Characteristics

In this study, two symmetric heptad repeat sequences were connected by a short loop, and the sequence was designed as XXRXXRRzzRRXXRXX-NH_2_ (X represents F, I, and L; zz represents GG or pG). The molecular mass of all peptides was consistent with theoretical values through measurement, demonstrating that the peptides were successfully synthesized. The hydrophobicity order of the designed peptides was as follows: IR_pG_ > FR_pG_ > IR_GG_ > FR_GG_ = LR_pG_ = LRα > LR_GG_ (Table 1). MALDI-TOF MS of the designed peptides is shown in Appendix A, and HPLC spectra of the designed peptides is shown in Appendix A.

### 2.2. Circular Dichroism (CD) Spectroscopy

The secondary structure of the designed peptides was detected using CD spectra in 10 mM phosphate-buffered saline (PBS), 50% trifluoroethanol (TFE), and 30 mM sodium dodecyl sulfate (SDS). In 10 mM PBS, the six designed peptides containing a central loop exhibited disordered conformations, and the control LRα showed a more ordered conformation. In 50% TFE (mimicking the hydrophobic environment of the microbial membrane) and 30 mM SDS (environment comparable to a negatively charged prokaryotic membrane) [15], FR_GG_, FR_pG_, IR_GG_, and IR_pG_ showed disordered conformations. LR_pG_ and LR_GG_ had a tendency to form an α-helical structure with negative peaks at approximately 205 nm and 220 nm. In contrast, although the hydrophobic and net charges of the control peptide LRα were consistent with LR_pG_, a typical α-helical structure with negative peaks at approximately 208 nm and 222 nm was observed (Figure 1). The results reveled that the central loop (pG or GG) disrupted the formation of the α-helical structure. The helical wheel projections of LR_GG_, LR_pG_, and LRα are shown in Appendix A.

### 2.3. Antimicrobial Activity

The antimicrobial activity of designed peptides is summarized in Table 2. The designed peptides showed better antimicrobial activity against Gram-negative bacteria than Gram-positive bacteria. For Gram-negative bacteria, LR_pG_ showed the best antimicrobial activity among the six designed peptides containing a central loop (LR_pG_ > LR_GG_ > FR_pG_ > FR_GG_ > IR_pG_ > IR_GG_), and the lowest geometric mean minimal inhibitory concentration (GM_MIC_) value of 3.5 μM. The control peptide LRα exhibited antimicrobial activity that was a little better than LR_pG_ with a GM_MIC_ value of 3 μM. For Gram-positive bacteria, peptides LR_ZZ_ and FR_ZZ_ had a relatively weaker antimicrobial effect compared to the control peptide LRα (Table 2).

### 2.4. Biocompatibility Assays

The hemolytic activity of designed peptides against human red blood cells (hRBCs) was determined in a range of 1–512 μM. The MHC_10_ values of IR_GG_, IR_pG_, FR_GG_, FR_pG_, LR_GG_, LR_pG_, and LRα were 512 μM, 512 μM, 128 μM, 128 μM, 256 μM, 256 μM, and 4 μM, respectively (Figure 2a). In general, the cytotoxicity of the designed peptides was consistent with the hemolytic activity. Later on, the murine macrophage cell line (RAW 264.7) and human embryonic kidney cells (HEK293T) were treated with 64 μM peptides, and the cell survival rate of RAW 264.7 cells was as follows: IR_pG_ (98%) > IR_GG_ (95%) > LR_pG_ (92%) > LR_GG_ (84%) > FR_pG_ (78%) > FR_GG_ (66%) (Figure 2b). Similarly, the cell survival rate of HEK293T cells was as follows: IR_pG_ (98%) > IR_GG_ (96%) > LR_pG_ (93%) > LR_GG_ (91%) > FR_pG_ (85%) > FR_GG_ (76%) (Appendix A). However, the cell survival rate of both cells treated with LRα was less than 10%. To determine the cell selectivity of the designed peptide, the therapeutic index (TI) (MHC_10_/GM_MIC_) of each peptide was calculated. Among all tested peptides, LR_pG_ showed the highest GM_TI_ (73.1) against Gram-negative bacteria (Table 2).

### 2.5. Condition Sensitivity Assays

These results demonstrated that LR_pG_ could maintain antimicrobial activity in different conditions. Therefore, LR_pG_ may have good potential in clinical application (Table 3 and Table 4).

### 2.6. Additive Effect of AMPs and Conventional Antibiotics

Based on the MICs of the designed peptides and antibiotics, the antimicrobial interactions of peptides and antibiotics were determined using a checkerboard assay. The combination of LR_pG_ and streptomycin exerted a synergistic effect against *Escherichia coli* ATCC25922. The fractional inhibitory concentration index (FICI) value was 0.5, while the combination of LR_pG_ with other conventional antibiotics produced an additive effect with FICI values from 0.625 to 1 (Table 5).

### 2.7. Antimicrobial Mechanism Study

#### 2.7.1. Outer Membrane Permeability Assay

Most AMPs kill bacteria by destroying the bacterial cell membrane. Since the outer membrane of the cell is negatively charged at physiological pH, the permeability of the cationic AMPs to the outer membrane is increased [21]. The permeability of peptides to the Gram-negative bacterial outer membrane can be reflected by *N*-phenyl-1-naphthylamine (NPN) uptake [15,17,21]. NPN is normally excluded from the outer membrane; however, once the outer membrane is permeabilized, NPN is taken up intracellularly and shows increased fluorescence. As shown in Figure 3, LR_pG_ and LRα were able to permeabilize the outer membrane of *E. coli* ATCC25922 at concentrations from 1 to 32 μM in a concentration-dependent manner. At peptide concentrations greater than 8 µM, the outer membrane permeability induced by LR_pG_ and LRα was over 75%. Moreover, the outer membrane permeability induced by LRα was slightly stronger than that induced by LR_pG_ at the same concentration.

#### 2.7.2. Inner Membrane Permeability Assay

The LR_pG_ permeability to the Gram-negative bacterial inner membrane was determined by measuring the cytoplasm β-galactosidase activity. LR_pG_ and LRα were able to induce an increase in the inner membrane permeability from 0 to 40 min. However, LRα displayed an optical density value at 420 nm (OD_420_) much higher than LR_pG_ at the same point in time. This phenomenon was different from the results of outer membrane permeability (Figure 4).

#### 2.7.3. Cytoplasmic Membrane Depolarization

In addition to inner and outer membrane permeability, changes in the plasma membrane potential energy of *E. coli* ATCC25922 were determined using 3,3’-dipropylthiadicarbocyanine (diSC_3_-5). The depolarization ability of cell membranes induced by LR_pG_ and LRα was dose- and time-dependent over 1500 s of measured time. This experiment showed that LRα had a faster and stronger depolarization ability than LR_pG_ (Figure 5).

#### 2.7.4. Scanning Electron Microscopy (SEM)

The cell morphology and membrane damage of *E. coli* ATCC25922 after LR_pG_ and melittin treatment were observed by SEM. In the control group, *E. coli* cells had a smooth and bright surface (Figure 6a), while those treated with peptides showed a rough surface with blebbing, as well as shrunken and destroyed shapes. The effect observed in the SEM micrographs was stronger in the cells treated with LR_pG_ than with the control (Figure 6b,c).

#### 2.7.5. DNA Binding Assay

The antibacterial effects of AMPs are exerted not only by acting on cell membranes, but also by binding to intracellular substances, such as DNA. Therefore, the ability to induce intracellular effects was measured using a DNA binding analysis. As shown in Figure 7, LR_pG_ and LRα exhibited binding ability at 64 μM and 16 μM, respectively. This experiment showed that LRα had a stronger DNA binding ability than LR_pG_.

#### 2.7.6. Lipopolysaccharide (LPS) Binding Assay

To detect the binding ability between peptides and LPS, the secondary structure of the peptides was measured using the CD spectrum in different concentrations of LPS. As shown in Figure 8, in 10 mM PBS without LPS solution, LR_pG_ exhibited disordered conformations and LRα exhibited a more ordered conformation. However, with the increase in LPS concentration, LR_pG_ and LRα tended to gradually form an α-helical structure. At the molar ratio peptide/LPS of 1:1, both LR_pG_ and LRα began forming an α-helical structure, indicating a similar binding ability to LPS.

#### 2.7.7. Limulus Amoebocyte Lysate (LAL) Assay

The chromogenic limulus amoebocyte lysate (LAL) assay was used to analyze the binding ability of the peptide to lipopolysaccharide (LPS). As shown in Figure 9, the activation of the LPS-mediated LAL coagulase was effectively inhibited by both LR_pG_ and LRα in a concentration-dependent manner. At low concentrations, the binding ability of LR_pG_ to LPS was much stronger than LRα, but the difference tended to be diminished with the increase in peptide concentration. 

#### 2.7.8. Endotoxin Neutralization Assay

To detect the inhibitory effect of LR_pG_ on the inflammatory response induced by LPS, the expression levels of two major inflammatory cytokines, tumor necrosis factor α (TNF-α) and nitric oxide (NO), in the macrophage supernatant were measured. Cells without stimulation and cells stimulated with only LPS (100 ng·mL^−1^) were used as negative and positive controls, respectively. As can be seen from Figure 10, LR_pG_ restrained the production of NO and TNF-α in a concentration-dependent manner and exhibited a significant inhibitory effect at 16 μM. The results showed that LR_pG_ had a good neutralization ability in LPS.

## 3. Discussion

Most AMPs have unique mechanisms of killing bacteria, mainly involving membrane breaking, which endows them with the potential to be alternatives to antibiotics [2]. However, the progress of natural AMPs for therapeutic application is seriously hindered by their inherent defects (such as manufacturing costs, cytotoxicity, poor stability etc.) [2,8]. To overcome these inherent shortcomings, in this study, six symmetric heptad repeat sequences were connected by a short loop, and the sequence was designed as XXRXXRRzzRRXXRXX-NH_2_ (X represents F, I, and L; zz represents GG or pG).

The results of MICs indicated that the designed peptides had good antibacterial activity toward Gram-negative bacteria (Table 2). Compared to the single membrane and thick peptidoglycan layer of Gram-positive bacteria, the double-membrane structure of Gram-negative bacteria represents a weaker barrier; thus, Gram-negative bacteria are more sensitive to AMPs [17,21,24]. The antibacterial activity of AMPs can be affected by many factors. Previous studies demonstrated that a proper positive charge (+6 to +7) was essential for antibacterial activity, but antibacterial activity was no longer increased when the positive charge of AMPs was beyond the threshold [17,20]. In this study, the positive charge of the designed peptides was set to +6. Based on their hydrophobicity, the theoretical antibacterial activity of the designed peptides would be IR_ZZ_ > FR_ZZ_ > LR_ZZ_ = LRα; however, the result was LRα > LR_ZZ_ > FR_ZZ_ > IR_ZZ_. Combined with the results of CD, although both LR_ZZ_ and LRα had the tendency to form an α-helical structure in membrane-mimetic environments, LRα’s tendency was stronger. One possible explanation is that the rigid loop (pG) of LR_ZZ_ disrupted the α-helical structure. Furthermore, we found that both FR_ZZ_ and IR_ZZ_ exhibited disordered conformations in membrane-mimetic environments. The result further confirmed the previous perspectives that the antibacterial activity of designed peptides is not only connected to the hydrophobicity of the peptides but also to the spatial stability of the secondary structure [25]. In addition, the antibacterial activity of XR_pG_ was higher than that of XR_GG_. Some papers confirmed that the rigid loop (pG), which promotes tightness in the hydrophobic center, has a better effect than the flexible loop (GG) [20].

However, the antimicrobial activity of AMPs is not the only criterion when evaluating their potential for clinical application. The hemolytic activity and cytotoxicity to mammalian cells are also major limiting factors [26,27]. Therefore, the cytotoxicity and hemolytic activity of the designed peptides were tested. At all examined concentrations, the designed peptides had much lower hemolysis and cytotoxicity (Figure 2), showing better cell selectivity compared to LRα. This result may be due to the insertion of the central loop as previous papers reported. The cell selectivity of XR_pG_ was better than that of XR_GG_, which may be due to the rigid pG having a better effect on promoting cell selectivity than flexible GG [20]. As shown in Table 2, LR_pG_ had a great balance between antibacterial activity and cytotoxicity, and it had higher cell selectivity.

In addition to the cell selectivity assays of AMPs, stability is also a major reason impeding AMPs from becoming therapeutic agents [2,28,29]. In this study, the effects of the physiological concentrations of salts, temperature, and pH on the antibacterial activity of the designed peptides were investigated. As shown in Table 3, in salt ions, the antimicrobial activity of LR_pG_ against *E. coli ATCC25922* was negligibly affected by all tested cations except for Na^+^. Previous reports suggested that a stable α-helical structure helps salt tolerance [30]. However, the insertion of the loop (pG) structure into the sequence template still resulted in good salt stability in this study. The result was due to the loop (pG) promoting hydrogen bonding and hydrophobic interactions, thereby achieving the effect of salt tolerance [31]. The antibacterial activity of LR_pG_ against *E. coli* ATCC25922 decreased because of the disruption of electrostatic interaction between LR_pG_ and the cell membrane in the presence of 150 mM Na^+^ [32]. For the clinical application of AMPs, thermal stability is very important since many foods and feed products need to be heated during processing [33]. LR_pG_ exerted good thermal stability (Table 4) and could maintain antibacterial activity in different pH conditions.

Furthermore, LR_pG_ exhibited a synergistic effect with streptomycin and additive effects with other antibiotics against *E. coli* ATCC25922 (Table 5). Some studies reported that the antibacterial effect of streptomycin and chloramphenicol involved acting on ribosomes, while ciprofloxacin acted on DNA gyrase, and cefotaxime acted on the cell membrane. In this study, LR_pG_ acted on cell membranes to increase membrane permeability and promote antibiotic entry into cells [34,35]. The combination of LR_pG_ and antibiotics improved the binding efficiency and ultimately enhanced the therapeutic efficacy. Therefore, LR_pG_ can serve as an additive to decrease the dose of antibiotics.

In this study, we also compared a natural AMP (melittin) to LR_pG_ in the GM_MIC_, MHC_10_, GM_SI_, and stability results. Although it exhibited a slightly better antibacterial activity than LR_pG_, melittin had a strong hemolysis effect. By calculating the GM_SI_ values (MHC_10_/GM_MIC_), we found that the GM_SI_ of LR_pG_ was 584 times higher than that of melittin. Furthermore, LR_pG_ exerted a good conditional stability and additive effects with conventional antibiotics. Based on the results above, LR_pG_ overcomes the main limiting factors of natural AMPs (such as melittin) and has the potential to develop into a therapeutic agent. Therefore, we further studied the antibacterial mechanism of LR_pG_. Previous studies demonstrated that cationic AMPs combine with anions on the surface of bacteria via electrostatic action, and their hydrophobic amino acids are inserted into the bacterial cell membrane. Once the critical concentration is exceeded, the membrane breaks, and this ultimately leads to death of the bacteria [21,22]. LR_pG_ firstly bound to the LPS of the outer membrane of *E. coli* ATCC25922 through electrostatic action (Figure 8), and then destroyed the cell membrane. Compared to the outer membrane, LR_pG_ had a much weaker permeability to the inner membrane than LRα (Figure 3 and Figure 4). This result might be due to the decreased helical tendency of the LR_pG_ as a result of the loop (pG), which was consistent with the previous papers showing that the helical tendency of AMPs was closely related to the permeability of bacterial inner membrane [20]. After breaking the impermeable barrier, LR_pG_ interacted with the cytoplasmic membrane, resulting in pore and ion channel formation (Figure 5). SEM results directly confirmed that LR_pG_ killed *E. coli* (Figure 6) by destroying the bacterial cell membranes and causing intracellular lysate leakage. In addition to the membrane mechanisms mentioned above, LR_pG_ can bind to intracellular material DNA at a concentration much higher than the MIC (Figure 7), suggesting that LR_pG_ cannot kill bacteria by DNA binding at the MIC. Furthermore, we found that the DNA binding ability of LR_pG_ was relatively weaker than that of LRα. One possible reason is that the α-helical forming ability of LR_pG_ was weaker than that of LRα. In short, LR_pG_ exerted its antibacterial effect by destroying the integrity of the bacterial cell membrane, causing leakage of the cellular content and binding to DNA through the cell membrane.

LPS is a crucial component of the outer membrane in Gram-negative bacteria [36]. Following bacterial death under the action of antibacterial agents, a large amount of LPS detaches from the cells and enters into the blood circulation of the body, activates the inflammatory signaling pathway, and releases inflammatory mediators such as NO and TNF-α [37,38]. Previous studies showed that positively charged amphipathic AMPs had a strong LPS neutralization ability and were ideal candidates for anti-inflammatory drugs [39,40]. The LPS-induced inflammatory mediators NO and TNF-α were remarkably inhibited by LR_pG_ in a concentration-dependent manner (Figure 10). Based on this study, LR_pG_ has the potential to develop into an anti-inflammatory agent by blocking LPS-mediated inflammatory mediators.

## 4. Materials and Methods 

### 4.1. Materials

*Escherichia coli* (*E. coli*) ATCC25922, *E. coli* K88, *Pseudomonas aeruginosa* (*P. aeruginosa*) ATCC27853, *Salmonella* Pullorum (*S.* Pullorum) NCTC5776, *Klebsiella pneumoniae* (*K. pneumoniae*) CMCC46117, *Staphylococcus aureus* (*S. aureus*) ATCC25923, *S. aureus *ATCC29213, methicillin-resistant *S. aureus* ATCC43300, and *Enterococcus faecalis* (*E. faecalis*) ATCC29212 were obtained from the pharmacology and toxicology laboratory of the College of Animal Science and Technology, Jilin Agricultural University (Changchun, China). HEK293T and RAW 264.7 cells were obtained from the College of Animal Science and Technology, Jilin Agricultural University (Changchun, China). The hRBCs were obtained from the Jilin Agricultural University Hospital.

Mueller–Hinton broth (MHB) powder and Mueller–Hinton agar (MHA) were purchased from GL Biochem (Shanghai, China). Bovine serum albumin (BSA), Triton X-100, and phosphate-buffered saline (PBS) solution were purchased from Solarbio (Beijing, China). *O*-nitrophenyl-β-d-galactopyranoside (ONPG), LPS from *E. coli* 0111 B4, diSC_3_-5, polymyxin B, 4-(2-hydroxyethyl) piperazine-1-ethanesulfonic acid (HEPES), and NPN were all purchased from Sigma-Aldrich (Shanghai, China). TransDetect Cell Counting Kit-8 (CCK-8) and Dulbecco’s modified Eagle’s medium with high glucose (DMEM) were purchased from TransGen Biotech (Beijing, China). Fetal bovine serum (FBS) was purchased from Gibco (Shanghai, China).

### 4.2. Synthesis and Sequence Analysis of Peptides

The designed peptides were sent to GL Biochem (Shanghai, China) for synthesis and purification. The purity of the peptide was determined by reverse-phase high-performance liquid chromatography on an analytical Kromasil C_18_ column (Beijing, China). The purity of the peptide was over 95% (RP-HPLC). ESI-MS analysis was used to measure the molecular weight of designed peptides, and values were very close to the calculated values. The primary physical and chemical parameters of the designed peptides were measured online at http://web.expasy.org/compute_pi/. The hydrophobicity and relative hydrophobic moments were measured online at http://heliquest.ipmc.cnrs.fr/.

### 4.3. CD Spectroscopy

The CD Spectroscopy of the designed peptides was detected on a J-810 spectropolarimeter (Jasco, Tokyo, Japan) at 25 °C. The final concentration of peptides in 10 mM PBS (pH 7.4), 50% TFE, and 30 mM SDS was 150 µM. The experimental parameters were as follows: 10 nm·min^−1^ scanning rate, 1 mm path length, and 190–250 nm wavelength range. Each peptide was scanned an average of three times. The collected data were transformed to mean residue ellipticity as follows:(1)$$\theta_{M}\;=\;\frac{\theta_{\mathrm{obs}}\;\times \;1000}{l\;\times \;c\;\times \;n}$$
where θ_M_ represents the mean residue ellipticity (deg·cm^2^·dmol^−1^), θ_obs_ represents the ellipticity correction of that measurement buffer at a given wavelength (mdeg), l represents the path length (cm), c represents the peptide concentration (mM), and n represents the number of amino residues.

### 4.4. MIC Measurements

The antimicrobial activity of the designed peptides was examined using the micro-dilution method described previously [29]. The bacteria were cultured for 16 h in MHB at 37 °C, and the microbial suspension was diluted to a final concentration of 10^5^ colony-forming units (CFU)/mL. Then, 50 µL of MHB containing different concentrations peptides (0.25–256 µM) was mixed with 50 µL of bacterial solution, and the mixture was added to a 96-well plate. After incubation for 16 h, MIC was examined by measuring the OD at 492 nm (Microplate reader, TECAN GENios F129004, Tecan, Salzburg, Austria). The assays were performed three times.

### 4.5. Biocompatibility Assays

The hemolytic activity was examined based on the method mentioned previously [22]. Briefly, the collected hRBCs were washed three times and diluted at 2% (*v/v*) with PBS (pH 7.4). Then, 100 µL of hRBCs were put into each tube including 100 µL of two-fold diluted peptides. The mixtures were centrifuged at 1000× *g* for 3 min. The OD value at 570 nm of the supernatant was measured, describing hemoglobin release (Microplate reader, TECAN GENios F129004, Tecan, Salzburg, Austria). For this assay, 1% Triton X-100-treated hRBCs were used as a positive control and untreated hRBCs were used as a negative control. MHC_10_ was defined as the peptide concentration that caused 10% hemolysis. The assays were performed three times.

The cytotoxicity was detected according to a CCK-8 assay as described previously. Briefly, RAW 264.7 and HEK293T cells were placed into 96-well plates at a density of 1.0–2.0 × 10^5^, and then cultured for 16 h under conditions of 5% CO_2_. Then, two-fold diluted peptides in DMEM were added to the plates. After 16 h of culturing, CCK-8 (10%, *v/v*) was put into each well containing cell culture, and mixtures were cultured for 2 h. The cytotoxicity assay was examined by measuring the OD of the mixtures at 450 nm (Microplate reader, TECAN GENios F129004, Tecan, Salzburg, Austria). The assays were performed three times.

### 4.6. Condition Sensitivity Assays

The salt, temperature, and pH sensitivities of designed peptides were determined by MIC assay as described previously [40]. For the salt sensitivity assay, peptides were diluted in deionized water containing different salts, and mixed with *E. coli* ATCC25922 (10^5^ CFU/mL). The final physiological concentrations of different salts were as follows: 150 mM NaCl, 4.5 mM KCl, 6 μM NH_4_Cl, 1 mM MgCl_2_, 8 μM ZnCl_2_, and 4 μM FeCl_3_. For temperature sensitivity, peptides were cooled on ice for 10 min, and incubated at 37 °C and 100 °C for 30 min. For pH sensitivity, peptides were treated with different pH for 1 h. Briefly, peptides were diluted in deionized water, and the pH of the solution was adjusted to 4.0, 6.0, 8.0, and 10.0 with HCl or NaOH. The subsequent steps were the same as for the MIC measurements. These tests were performed three times.

### 4.7. Synergy with Conventional Antibiotics

The combined antibacterial effect of peptides with other antibiotics was evaluated by checkerboard assay as described previously [31]. Briefly, peptides and antibiotics diluted two-fold were prepared at concentrations ranging from 1/8× MIC to 2× MIC. Subsequently, each peptide at the same concentration was added to the longitudinal column of wells, and each antibiotic at the same concentration was added to the horizontal row of wells. A suspension at 10^5^ CFU/mL was put into each well for 16 h at 37 °C. Each assay was performed three times. The fractional inhibitory concentration (FIC) index was calculated as follows:(2)$$\mathrm{FIC}\;=\;\frac{\mathrm{MIC}\;\mathrm{of}\;\mathrm{drug}\;A\;\mathrm{in}\;\mathrm{combination}}{\mathrm{MIC}\;\mathrm{of}\;\mathrm{drug}\;A\;\mathrm{alone}}+\frac{\mathrm{MIC}\;\mathrm{of}\;\mathrm{drug}\;B\;\mathrm{in}\;\mathrm{combination}}{\mathrm{MIC}\;\mathrm{of}\;\mathrm{drug}\;B\;\mathrm{alone}}$$
where FIC ≤ 0.5 denotes synergy, 0.5 < FIC ≤ 1.0 denotes additive, and 1.0 < FIC ≤ 4.0 denotes indifferent.

### 4.8. Outer Membrane Permeability Assay

The outer membrane permeability was examined using fluorescent dye NPN according to the method mentioned previously [17,21,41]. Briefly, *E. coli* ATCC25922 cells were washed three times and diluted to 10^5^ CFU/mL in 5 mM HEPES buffer (pH 7.4, containing 5 mM glucose). Then, the final concentration of 10 μM NPN was added to the suspension, and incubated at room temperature in the dark for 30 min. An equal volume of suspension and peptides at various concentrations were mixed in a black 96-well plate. The fluorescence intensities of the samples were measured (emission λ = 420 nm, excitation λ = 350 nm) with an F4500 fluorescence spectrophotometer (Hitachi, Tokyo, Japan). Polymyxin B (10 µg/mL) with strong outer membrane permeability was used as a positive control. Values were converted to percentage NPN uptake using the following equation:(3)$$\mathrm{NPN}\;\mathrm{uptake}\;(\%)\;=\;\frac{F_{\mathrm{obs}}-F_{0}}{F_{100}-F_{0}}\times 100$$
where F_obs_ is the observed fluorescence of NPN with *E. coli* ATCC25922 cells at a given peptide concentration, F_0_ is the initial fluorescence of NPN with *E. coli* ATCC25922 cells, and F_100_ is the fluorescence of NPN with *E. coli* ATCC25922 cells with the addition of 10 µg/mL polymyxin B.

### 4.9. Inner Membrane Permeability Assay

The inner membrane permeabilization was determined by measuring the cytoplasm β-galactosidase activity according to the method mentioned previously [21]. In short, an equal volume of the suspension of *E. coli* ATCC25922 cells was washed three times and diluted to an OD_600_ of 0.05 with 10 mM PBS (containing 1.5 mM ONPG); then, the suspension was put into each well of a 96-well plate and cultured with 0.5× MIC to 4× MIC peptides at 37 °C. The OD at 420 nm (Microplate reader, TECAN GENios F129004, Tecan, Salzburg, Austria) was recorded for 40 min every 5 min eight times. The method was used to detect the permeabilization of the inner membrane through the reflection of the ONPG influx into the cells.

### 4.10. Cytoplasmic Membrane Depolarization Assay

DiSC_3_-5, a membrane potential-sensitive fluorescent dye, was used to assess the ability of the peptides to disrupt the cytoplasmic membrane as previously described, [42]. Briefly, *E. coli* ATCC25922 in the mid-log phase was washed three times and diluted to an OD_600_ of 0.05 with 5 mM HEPES buffer (containing 20 mM glucose and 100 mM KCl); then, the mixture of *E. coli* ATCC25922 and 0.5× MIC to 4× MIC peptides was mixed with diSC_3_-5 (final concentration of 0.4 μM) at 37 °C for 1 h. The fluorescence changes from 0 to 1500 s were recorded at the emission wavelength (λ = 670 nm) and excitation wavelength (λ = 622 nm) (Fluorescence Spectrophotometer, F4500, Hitachi, Tokyo, Japan).

### 4.11. Scanning Electron Microscopy

*E. coli* ATCC25922 cells in the mid-log phase were washed three times and diluted to 10^7^ CFU/mL in 10 mM PBS. The bacterial suspension and peptide (1× MIC) were added to 24-well plates containing polylysine-treated glass slides. The bacteria untreated by peptide were used as a control. After incubation, the bacterial suspension was removed, and the treatment method with polylysine-glass slides was repeated. Then, the polylysine-treated glass slides were fixed with glutaraldehyde (2.5%) at 4 °C for 12 h and dehydrated with a series of graded ethanol solutions for 20 min. The polylysine-treated glass slides were transferred into a mixture (*v*:*v* = 1:1) of pure alcohol and tertiary butanol for 15 min, and then placed into pure tertiary butanol for 15 min. Finally, the samples were treated using a critical point dryer with liquid CO_2_, and then scanned by a S-3400N SEM (Hitachi, Tokyo, Japan) after coating with gold-palladium.

### 4.12. DNA Binding Assay

The DNA binding assays were determined according to gel retardation experiments described previously [17]. Briefly, 400 ng of genomic DNA (*E. coli* ATCC25922) was mixed with 0.5 to 256 µM peptide in a binding buffer (50 µg/mL BSA, 1 mM ethylene diamine tetraacetic acid (EDTA), 20 mM KCl, 10 mM Tris-HCl (pH = 8.0), 5% glycerol, and 1 mM dithiothreitol) for 60 min at 37 °C. Subsequently, the samples were examined by 1% agarose gel electrophoresis.

### 4.13. LPS Binding Assay

The ability of peptides to bind LPS was detected on a J-810 spectropolarimeter at 25 °C. Briefly, concentration of LPS ranged from 37.5–600 µM, diluted using the double dilution method. The final concentration of peptides in the LPS solution was 150 µM. The experimental parameters were the same as in Section 4.3.

### 4.14. LAL Assay

Peptide neutralization was evaluated using a quantitative chromogenic limulus amoebocyte lysate (LAL) assay (Xiamen, China) [43]. Briefly, 1 endotoxin unit (EU) of LPS was incubated with two-fold serial dilutions of peptides at 37 °C for 60 min; then, 100 µL of LAL reagent was added to the mixtures. After additional incubation for 15 min at 37 °C, 100 µL of chromogenic substrate solution was put into the tubes. Subsequently, the mixtures were incubated for another 10 min at 37 °C before adding 500 µL of azo reagent solutions (1, 2, and 3) in turn. The data were examined by measuring the OD value at 545 nm (Microplate reader, TECAN GENios F129004, Tecan, Salzburg, Austria).

### 4.15. Endotoxin Neutralization Assay

RAW 264.7 cells were put into 24-well plates at a density of 1.0–2.0 × 10^5^ cells/well and stimulated with LPS (100 ng/mL) with (1–64 μM) or without peptide. After incubating for 24 h at 37 °C in 5% CO_2_, the supernatants were collected for analysis of the NO production using Griess reagent and TNF-α levels using ELISA based on the manufacturer’s protocol. The cells treated with LPS only and untreated cells were set as the positive and negative controls, respectively.

### 4.16. Statistical Analysis

ANOVA was used to analyze the data through the SPSS 18.0 software (IBM, Chicago, IL, USA). Quantitative data were expressed as the mean ± standard deviation. A *p*-value < 0.01 was considered to have statistical significance.

## 5. Conclusions

In this study, we designed peptides based on the sequence template XXRXXRRzzRRXXRXX-NH_2_, where X represents a hydrophobic amino acid (F, I, and L) and zz represents the loop (GG or pG). The designed peptides effectively enhanced cell selectivity because of the introduction of the loop (GG or pG). Among them, LR_pG_ (X: L, zz: pG) showed the highest average therapeutic index (GM_TI_ = 73.1), with better conditional stability, and it had an additive effect with conventional antibiotics. The antibacterial mechanism study demonstrated that LR_pG_ first destroyed the integrity of the bacterial cell membrane, caused leakage of the cellular content, then entered the cell and bound to the DNA, and ultimately led to bacterial death. Furthermore, LR_pG_ also had an obvious impact on anti-inflammatory ability. These results indicated that the design and optimization of natural AMPs in this study will be helpful in improving the clinical application of AMPs.

## Figures and Tables

**Figure 1 ijms-21-01140-f001:**
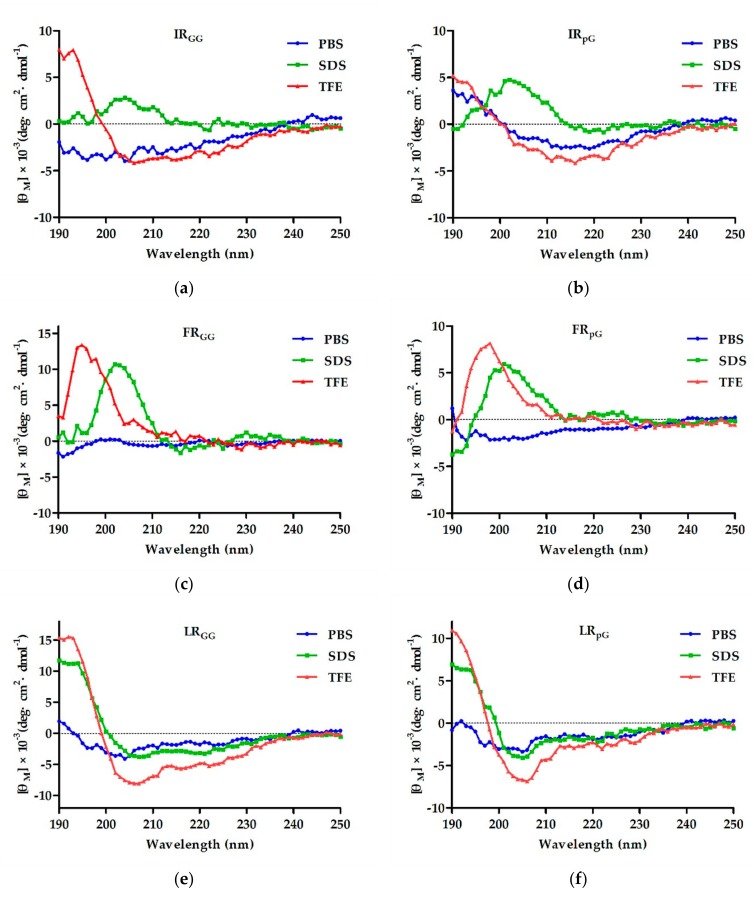
The circular dichroism (CD) spectra of (**a**) IR_GG_, (**b**) IR_pG_, (**c**) FR_GG_, (**d**) FR_pG_, (**e**) LR_GG_, (**f**) LR_pG_, and (**g**) LRα were dissolved in 10 mM phosphate-buffered saline (PBS) (**blue**), 30 mM sodium dodecyl sulfate (SDS) (**green**), and 50% trifluoroethanol (TFE) (**red**). The average value after three scans of every sample is shown. The CD spectrum of the buffer was subtracted.

**Figure 2 ijms-21-01140-f002:**
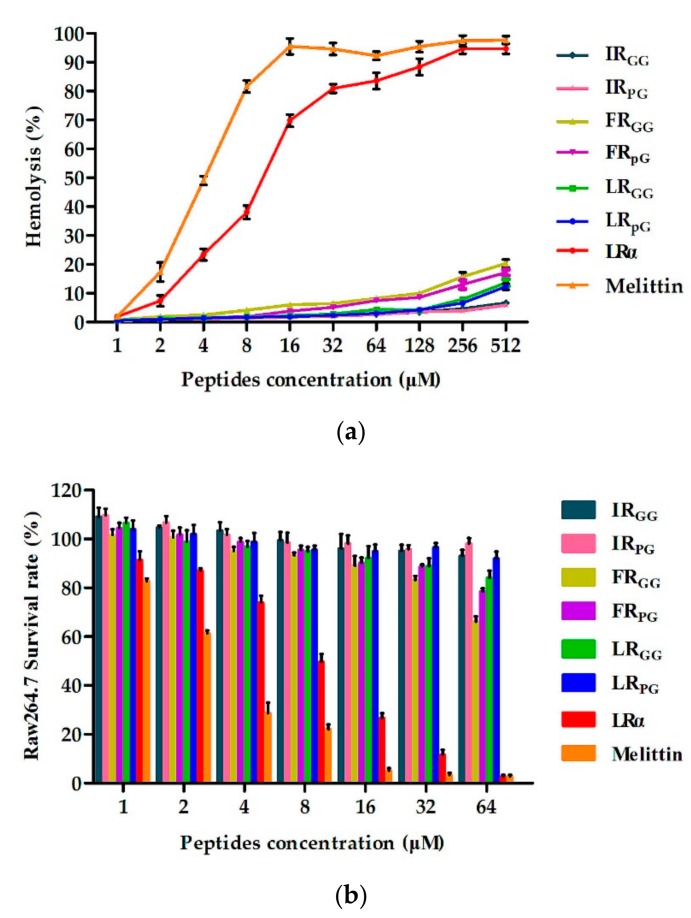
(**a**) Hemolytic activity of designed peptides against human red blood cells (hRBCs); (**b**) cytotoxicity of the designed peptides against RAW 264.7 cells. The diagrams are based on at least three independent experiments.

**Figure 3 ijms-21-01140-f003:**
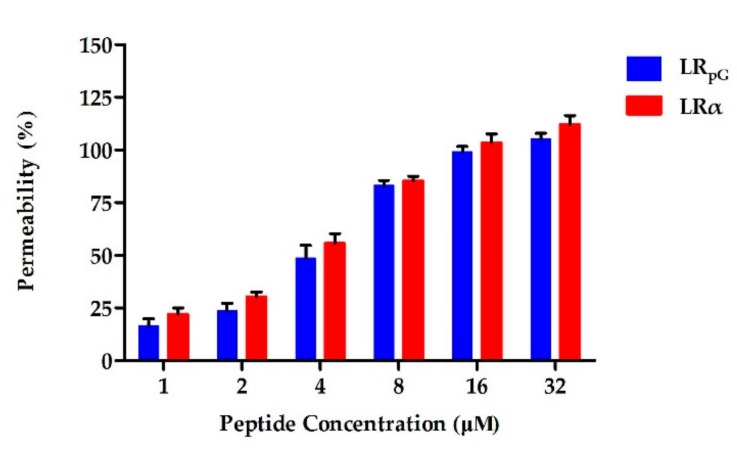
Outer membrane permeability of LR_pG_ and LRα at the concentrations from 1 to 32 μM. The diagram is based on at least three independent experiments.

**Figure 4 ijms-21-01140-f004:**
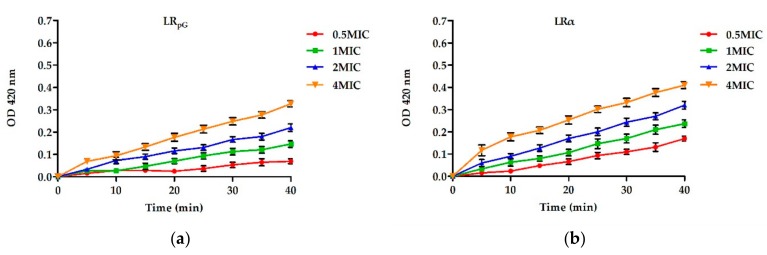
Inner membrane permeability of (**a**) LR_pG_ and (**b**) LRα at different concentrations. The diagrams are based on at least three independent experiments.

**Figure 5 ijms-21-01140-f005:**
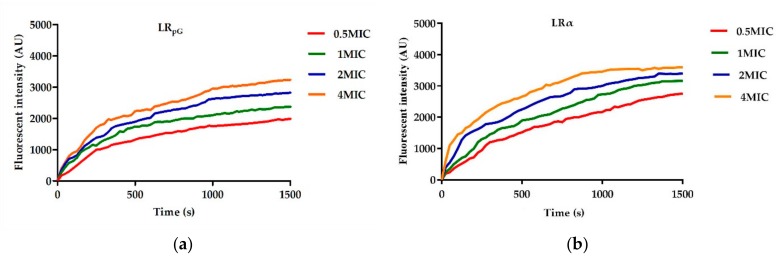
Cytoplasmic membrane potential variation of *E. coli* ATCC25922 treated with (**a**) LR_pG_ and (**b**) LRα at different levels of concentration. The diagram is based on at least three independent experiments.

**Figure 6 ijms-21-01140-f006:**

SEM micrographs of *E. coli* ATCC25922: (**a**) control; (**b**) LR_pG_-treated; (**c**) LRα-treated.

**Figure 7 ijms-21-01140-f007:**
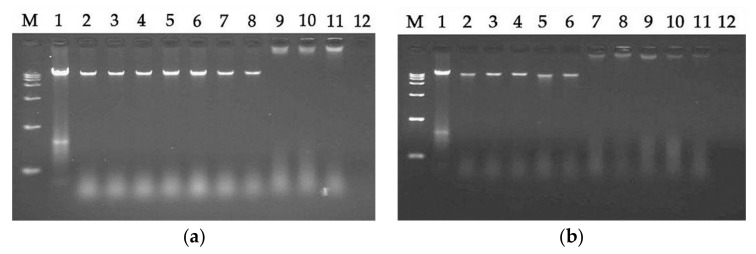
A gel retardation experiment was used to measure the DNA binding assay: (**a**) LR_pG_-treated; (**b**) LRα-treated. M: DNA marker alone; 1: genomic DNA alone; 2: with 0.5 μM peptide; 3: with 1 μM peptide; 4: with 2 μM peptide; 5: with 4 μM peptide; 6: with 8 μM peptide; 7: with 16 μM peptide; 8: with 32 μM peptide; 9: with 64 μM peptide; 10: with 128 μM peptide; 11: with 256 μM peptide; 12: 256 μM peptide alone.

**Figure 8 ijms-21-01140-f008:**
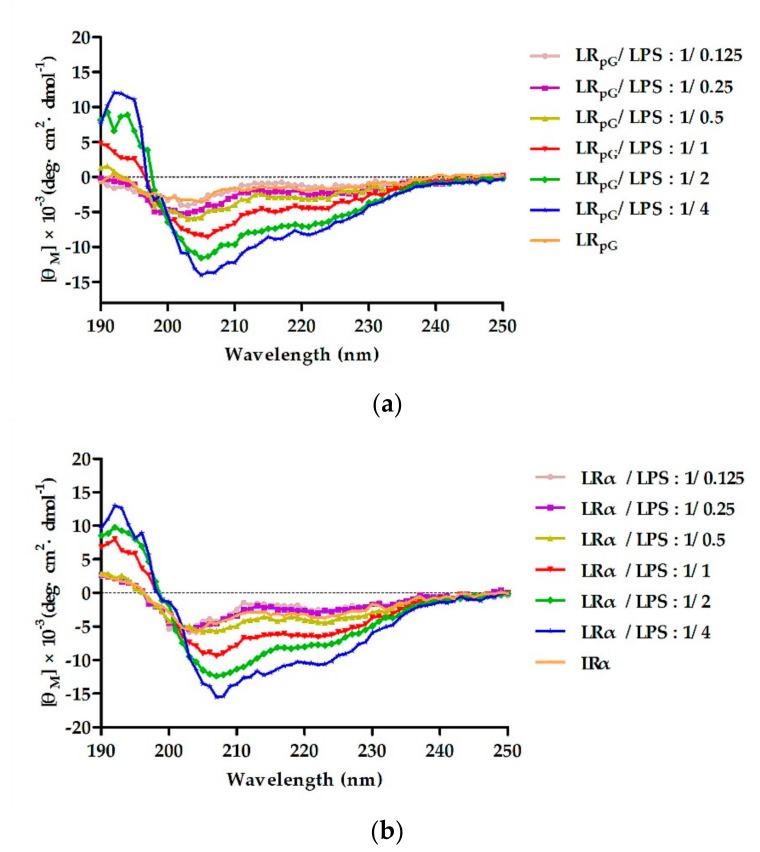
The circular dichroism (CD) spectra of (**a**) LR_pG_ and (**b**) LRα were dissolved in different concentrations of lipopolysaccharide (LPS), and the peptide/LPS molar ratio ranged from 1/0.125 to 1/4. The average value after three scans of every sample is shown. The CD spectrum of LPS was subtracted.

**Figure 9 ijms-21-01140-f009:**
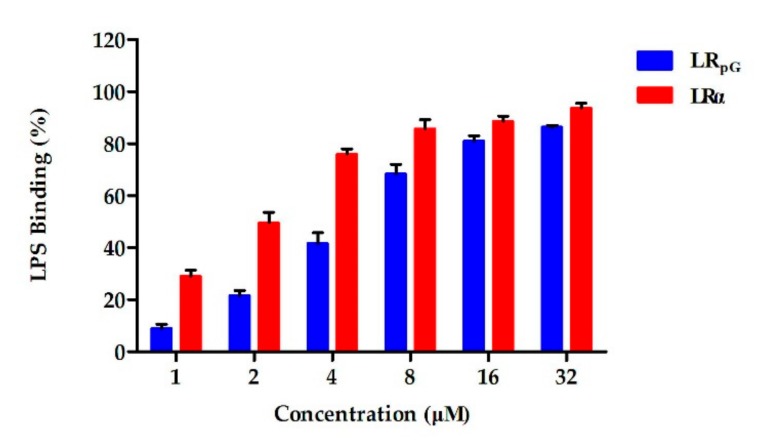
The binding ability of LR_pG_ and LRα to lipopolysaccharide (LPS), determined by limulus amoebocyte lysate (LAL) assay. The diagram is based on at least three independent experiments.

**Figure 10 ijms-21-01140-f010:**
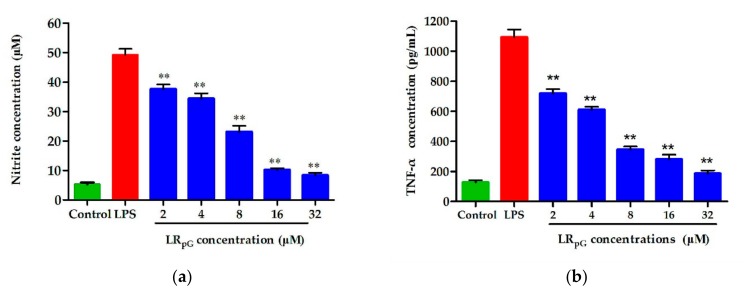
Effects of LR_pG_ on the production of (**a**) nitric oxide (NO) and (**b**) tumor necrosis factor α (TNF-α) in LPS-stimulated RAW 264.7 cells. The diagrams are based on the average of three independent experiments. ** *p* < 0.01, compared to the positive group (only LPS-stimulated).

**Table 1 ijms-21-01140-t001:** Peptides and their key physicochemical parameters.

Peptides	Sequence	Theoretical MW	Measured MW ^1^	Net Charge	H ^2^	µHre ^3^
IR_GG_	IIRIIRRGGRRIIRII-NH_2_	1974.52	1973.5	6	0.521	0.770
IR_pG_	IIRIIRRp ^4^ GRRIIRII-NH_2_	2014.58	2013.8	6	0.566	0.800
FR_GG_	FFRFFRRGGRRFFRFF-NH_2_	2246.66	2245.68	6	0.516	0.767
FR_pG_	FFRFFRRp ^4^ GRRFFRFF-NH_2_	2286.72	2285.74	6	0.561	0.797
LR_GG_	LLRLLRRGGRRLLRLL-NH_2_	1974.52	1973.51	6	0.471	0.743
LR_pG_	LLRLLRRp ^4^ GRRLLRLL-NH_2_	2014.58	2013.58	6	0.516	0.773
LRα	GLRLLRRLLRRLLRLp ^4^ -NH_2_	2014.58	2013.58	6	0.516	0.735

^1^ Molecular weight (MW) was determined by mass spectrometry (MS). ^2^ The mean hydrophobicity (H) is the total hydrophobicity (sum of all residue hydrophobicity indices) divided by the number of residues. ^3^ The relative hydrophobic moment of a peptide is its hydrophobic moment relative to that of a perfectly amphiphilic peptide. ^4^ Lowercase letters indicate the d-enantiomer of the proline.

**Table 2 ijms-21-01140-t002:** Minimum inhibitory concentrations (MICs) (µM) and therapeutic index (TI) of the designed peptides.

Bacterial Species		IR_GG_	IR_pG_	FR_GG_	FR_pG_	LR_GG_	LR_pG_	LRα	Melittin
MIC (µM)									
	Gram (−)								
	*Escherichia coli* ATCC25922	8	4	8	8	2	2	2	1
	*E. coli* K88	16	16	16	8	4	2	4	2
	*Salmonella* Pullorum NCTC5776	32	16	16	16	4	4	2	2
	*Klebsiella pneumonia* CMCC46117	32	32	16	16	8	4	4	4
	*Pseudomonas aeruginosa* ATCC27853	32	32	32	32	8	8	4	2
	Gram (+)								
	*Staphylococcus aureus* ATCC25923	>128	>128	32	16	16	16	4	2
	*S. aureus* ATCC29213	>128	>128	16	16	16	16	8	1
	*S. aureus* ATCC43300	>128	>128	32	32	32	16	8	2
	*Enterococcus faecalis* ATCC29212	>128	>128	32	32	32	32	16	1
GM ^1^ (µM)									
	Gram (−)	21.1	16	16	13.9	4.6	3.5	3	2
	Gram (+)	256	256	26.9	22.6	22.6	19.0	8	1.4
	MHC_10_ ^2^ (µM)	512	512	128	128	256	256	4	0.25
TI ^3^									
	TI (−)	24.2	32	8	9.2	55.6	73.1	1.3	0.125
	TI (+)	2	2	4.8	5.7	11.2	13.5	0.5	0.178

^1^ Geometric mean (GM) of MICs for the nine strains was measured. A value of 256 μM was used for the TI when no antimicrobial activity was examined at 128 μM. ^2^ Minimum hemolysis concentration 10% (MHC_10_) was characterized as the lowest concentration inducing 10% hemolysis. ^3^ TI was calculated as MHC_10_/GM.

**Table 3 ijms-21-01140-t003:** The MICs (µM) of the designed peptides against *E. coli* ATCC25922 in the presence of physiological salts.

Peptides	Control ^1^	Physiological Salts ^2^					
		NaCl	KCl	NH_4_Cl	MgCl_2_	ZnCl_2_	FeCl_3_
IR_GG_	8	64	8	8	16	16	8
IR_pG_	4	32	4	4	8	4	4
FR_GG_	8	64	8	8	16	16	8
FR_pG_	8	32	8	8	8	8	8
LR_GG_	2	8	2	2	4	2	2
LR_pG_	2	8	2	2	2	2	2
LRα	2	8	2	2	2	2	2
Melittin	1	4	1	1	1	1	1

^1^ The control MICs were determined in Mueller–Hinton broth MHB medium without physiological salts. ^2^ The final physiological concentrations of NaCl, KCl, NH_4_Cl, MgCl_2_, ZnCl_2_, and FeCl_3_ were 150 mM, 4.5 mM, 6 μM, 1 mM, 8 μM, and 4 μM, respectively. The data are based on at least three independent experiments.

**Table 4 ijms-21-01140-t004:** The MICs (µM) of the designed peptides against *E. coli* ATCC25922 after temperature and pH treatment.

Peptides	Control (pH 7)	Temperature			pH			
		0 °C	37 °C	100 °C	pH 4	pH 6	pH 8	pH 10
IR_GG_	8	8	8	8	16	8	8	16
IR_pG_	4	4	4	4	16	4	8	16
FR_GG_	8	8	8	8	32	8	8	32
FR_pG_	8	8	8	8	16	8	8	32
LR_GG_	2	2	2	2	2	2	2	4
LR_pG_	2	2	2	2	2	2	2	4
LRα	2	2	2	2	2	2	2	4
Melittin	1	1	1	1	2	1	1	2

**Table 5 ijms-21-01140-t005:** The fractional inhibitory concentration index (FICI) ^1^ for LR_pG_ in combination with conventional antibiotics against *E. coli* ATCC25922.

Peptide	Streptomycin ^2^	Ciprofloxacin ^2^	Chloramphenicol ^2^	Cefotaxime ^2^
LR_pG_	0.5	0.625	0.75	1

^1^ FICI ≤ 0.5 denotes synergy, 0.5 < FICI ≤ 1.0 denotes additive, and 1.0 < FICI ≤ 4.0 denotes indifferent. ^2^ The MICs of streptomycin, ciprofloxacin, chloramphenicol, and cefotaxime against *E. coli* ATCC25922 were 2 μM, 8 μM, 8 μM, and 2 μM, respectively.

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
