# Peer review of "High Cell Selectivity and Bactericidal Mechanism of Symmetric Peptides Centered on d-Pro–Gly Pairs"

_ijms, 2020, doi:10.3390/ijms21031140_

Round 1

Reviewer 1 Report

Comments to the authors

The new version of the manuscript sent by Jia B et al (now named as “High cell selectivity and bactericidal mechanism of symmetric peptides centered on D-Pro-Gly pairs”) has improved its quality through the changes done by the authors. The English language and style have been improved  by MDPI correction of the spelling, punctuation and mistakes. Furthermore, some additional experiments (e.g. circular dichroism in the presence of LPS) have been added as suggested. The authors have also changed the control to LRα, a more related peptide that allows the comparison of the results obtained for the designed peptides. In general, the authors have answered all the comments of my previous report. 

At this point, I would only ask for the addition of some sentences in the discussion in order to highlight the results obtained (e.g. compare the MIC values with the one of natural AMPs in order to know the relevance of the values obtained, see comment 21), the checking of the CD calculations (see comments 10, 12, 14 and 15) and some minor changes (see list below).

Introduction 

First paragraph. First sentence is too long. I recommend to split it in two shorter sentences. First paragraph. First sentence. Add “in” (“With the widespread use of antibiotics in clinical practice in recent years [...]) 3rd paragraph. “In this study, two symmetric heptad repeat sequences were conjugated with the loop”. I wouldn ́t use the term “conjugation” since the chemical meaning is different to the one described by the authors. Maybe the sentence “In this study, two symmetric heptad repeat sequences connected by a short loop” fits better the description of the peptide structures.  3rd paragraph. “In addition, the deeper insertion of membranes and stronger membrane destruction capabilities of AMP was promoted by Arg (R), Arg was selectively introduced into the heptad repeat sequence because the guanidine side chain and the phosphate moiety of the two lipid head group form strong bidentate H-bonds compared to Lys (K) [20]”. This sentence is too long. I recommend to split it in two shorter sentences.   Last sentence of the introduction explains the aim of the work but not the results. In order to make more attractive the introduction, I would add some details about the results obtained for LRpG (as shown in the abstract) as a final sentence. 

Results

2.1. Peptide Design and Characteristics

6. I wouldn´t use the term “conjugation” since the chemical meaning is different to the one described by the authors. Maybe the sentence “In this study, two symmetric heptad repeat sequences connected by a short loop” fits better the description of the peptide structures.

7. “Demonstrating that the peptide had been successfully synthesized”. Change “peptide” to “peptides”.

2.2. Circular Dichroism (CD) Spectroscopy

8. In PBS, the control doesn´t exhibit a spectrum corresponding to random coil since we cannot observe a minimum around 200nm. It seems that it exhibits the alpha helical typical spectrum with a deviation of the peak corresponding to 208nm. As it is difficult to know why this happens, I would be conservative in the explanation of the effect saying that the control shows a more ordered conformation. 

9. I cannot see the “abnormal peaks” described in “In 50% TFE (mimicking the hydrophobic environment of the microbial membrane) and 30 mM SDS environments (negatively charged prokaryotic membrane comparable environment) [15], FRGG, FRpG, IRGG and IRpG showed disordered conformations with abnormal peaks”.

10. In the CD experiments, has the baseline for the buffer been subtracted to the spectra?

11. Figure 1 (e). In  the title of the graph IRα has to be changed to LRα

12. Could the authors check the calculations for the ellipticity showed in the graphs? The signal is too low and I suspect that a factor of 10 could be missed in the calculations. 

2.7.4. Scanning Electron Microscopy (SEM)

13. I would add that the effect observed in the SEM micrographs is stronger in the cells treated with LRpG than with the control (as could be observed in the images).

2.7.6. LPS Binding Assay

14. Has the baseline for the buffer been subtracted to the spectra?

15. As LPS shows high absorption, the LPS spectra have to be subtracted to the LPS+peptide mixture as in Avitabile C, D'Andrea LD, Romanelli A. Circular Dichroism studies on the interactions of antimicrobial peptides with bacterial cells. Sci Rep. 2014. Has this subtraction been done?

16. Adding the ratio peptide:LPS in the legend is a common practice when you use the CD with LPS (see Avitabile C, D'Andrea LD, Romanelli A. Circular Dichroism studies on the interactions of antimicrobial peptides with bacterial cells. Sci Rep. 2014. )

Discussion

17. Change two to six : “In order to overcome these inherent shortcomings, in this study, two symmetric heptad repeat sequences were conjugated with a loop [...]” 

18. Same sentence. Change “conjugated” to “connected by”.

19. Reconsider the statement: “Gram-negative bacteria are more sensitive to AMPs, because their cell walls are thinner and the outermost layer is LPS compared to Gram-positive bacteria [24]”. Gram-negative are not more sensitive, in fact there are more AMPs being specific for Gram-positive bacteria as seen in Malanovic N, Lohner K. Antimicrobial Peptides Targeting Gram-Positive Bacteria. Pharmaceuticals (Basel). 2016 

20. Line 269: “promo ting”

21. The authors have used a more appropriate control for comparison of the results obtained with the designed peptides, but here in the discussion it would be useful to compare the results (MIC, hemolytic effect, stability) with a natural AMP as melittin. I would introduce some sentences comparing these values in order to know the relevance of the results obtained with LRpG.

Methods

22. Salt sensitivity assay. To test the pH effect, the buffer solution was adjusted. What is the composition of this buffer? 

Author Response

Response to Reviewer 1 Comments

    We are truly grateful to your comments and thoughtful suggestions, those comments are valuable for revising and improving our paper. We have studied comments carefully and have made correction which we hope meet with approval. Revised portion are marked in red in the manuscript. The responds to the comments are as flowing:

Introduction

Point 1: First paragraph. First sentence is too long. I recommend to split it in two shorter sentences. 

Response 1: We have changed the sentence to “With the widespread use of antibiotics in clinical practice in recent years, the resistance rate of bacteria to traditional antibiotics has been increasing. The development of antibacterial reagents with new antibacterial mechanisms are imminent.” (shown in Line 33-35)

Point 2: First paragraph. First sentence. Add “in” (“With the widespread use of antibiotics in clinical practice in recent years [...]) 

Response 2: We have revised the mistake (shown in Line 33).

Point 3: 3rd paragraph. “In this study, two symmetric heptad repeat sequences were conjugated with the loop”. I wouldn ́t use the term “conjugation” since the chemical meaning is different to the one described by the authors. Maybe the sentence “In this study, two symmetric heptad repeat sequences connected by a short loop” fits better the description of the peptide structures. 

Response 3: We have revised the manuscript according to your comments and suggestions (shown in Line 53).

Point 4: 3rd paragraph. “In addition, the deeper insertion of membranes and stronger membrane destruction capabilities of AMP was promoted by Arg (R), Arg was selectively introduced into the heptad repeat sequence because the guanidine side chain and the phosphate moiety of the two lipid head group form strong bidentate H-bonds compared to Lys (K) [20]”. This sentence is too long. I recommend to split it in two shorter sentences. 

Response 4: We have changed the sentence to “In addition, Arg (R) was selectively introduced into the sequence template. Due to the guanidine side chain of R and the phosphate moiety of the two lipid head groups can form strong bidentate H-bonds, the electrostatic action between the peptides and anionic bacterial membrane can be enhanced.” (shown in Line 58-62)

Point 5:  Last sentence of the introduction explains the aim of the work but not the results. In order to make more attractive the introduction, I would add some details about the results obtained for LRpG (as shown in the abstract) as a final sentence. 

Response 5: We have changed last sentences of the introduction to “By this method of designing and optimizing AMP, we obtained LRpG (X: L, zz: pG) with high cell selectivity and good conditional stability. The antibacterial mechanism study demonstrated that LRpG played the role of anti-Gram-negative bacteria through destroying cell membrane”. (shown in Line 67-70)

Results

 2.1. Peptide Design and Characteristics.

Point 6: I wouldn´t use the term “conjugation” since the chemical meaning is different to the one described by the authors. Maybe the sentence “In this study, two symmetric heptad repeat sequences connected by a short loop” fits better the description of the peptide structures.

Response 6: We have revised the manuscript according to your comments and suggestions (shown in Line 73).

Point 7: “Demonstrating that the peptide had been successfully synthesized”. Change “peptide” to “peptides”

Response 7: We have revised the mistake (shown in Line 76).

2.2. Circular Dichroism (CD) Spectroscopy

Point 8: In PBS, the control doesn´t exhibit a spectrum corresponding to random coil since we cannot observe a minimum around 200nm. It seems that it exhibits the alpha helical typical spectrum with a deviation of the peak corresponding to 208nm. As it is difficult to know why this happens, I would be conservative in the explanation of the effect saying that the control shows a more ordered conformation. 

Response 8: We have revised the manuscript according to your comments and suggestions (shown in Line 88).

Point 9: I cannot see the “abnormal peaks” described in “In 50% TFE (mimicking the hydrophobic environment of the microbial membrane) and 30 mM SDS environments (negatively charged prokaryotic membrane comparable environment) [15], FRGG, FRpG, IRGG and IRpG showed disordered conformations with abnormal peaks”.

Response 9: We have removed “with abnormal peaks” in manuscript (shown in Line 90-91).

Point 10: In the CD experiments, has the baseline for the buffer been subtracted to the spectra?

Response 10: We are very sorry for our negligence, the baseline for the buffer has been subtracted to the spectra (shown in Figure 1).

Point 11: Figure 1 (e). In the title of the graph IRα has to be changed to LRα.

Response 11: We have revised the mistake (shown in Figure 1).

Point 12: Could the authors check the calculations for the ellipticity showed in the graphs? The signal is too low and I suspect that a factor of 10 could be missed in the calculations. 

Response 12: We are very sorry for the wrong calculations of the mean residue ellipticity.

    We have recalculated the mean residue ellipticity according to the following equation: θM = (θobs × 1000)/(l × c × n), where θM represents to the mean residue ellipticity (deg·cm2·dmol−1), θobs represents ellipticity correction of that measurement buffer at a given wavelength (mdeg), l represents the path length (cm), c represents the peptide concentration (mM), and n represents the number of amino residues. (shown in Figure 1)

    This calculation equation has been added to Material Method 4.3.

2.7.4. Scanning Electron Microscopy (SEM)

Point 13: I would add that the effect observed in the SEM micrographs is stronger in the cells treated with LRpG than with the control (as could be observed in the images).

Response 13: We have revised the manuscript according to your comments and suggestions (shown in 183-184).

2.7.6. LPS Binding Assay

Point 14: Has the baseline for the buffer been subtracted to the spectra?

Response 14: We are very sorry for our negligence, the baseline for the buffer has been subtracted to the spectra (shown in Figure 8).

Point 15: As LPS shows high absorption, the LPS spectra have to be subtracted to the LPS+peptide mixture as in Avitabile C, D'Andrea LD, Romanelli A. Circular Dichroism studies on the interactions of antimicrobial peptides with bacterial cells. Sci Rep. 2014. Has this subtraction been done?

Response 15: We are very sorry for our negligence, the LPS spectra has been subtracted to the LPS+peptide mixture spectra (shown in Figure 8).

Point 16: Adding the ratio peptide:LPS in the legend is a common practice when you use the CD with LPS (see Avitabile C, D'Andrea LD, Romanelli A. Circular Dichroism studies on the interactions of antimicrobial peptides with bacterial cells. Sci Rep. 2014. )

Response 16: Thanks for your comments and suggestions, we have improved the experimental method according to the above reference, and peptide/LPS molar ratio was set from 1/0.125 to 1/4.

    We have recalculated the mean residue ellipticity according to the following equation: θM = (θobs × 1000)/(l × c × n), where θM represents to the mean residue ellipticity (deg·cm2·dmol−1), θobs represents ellipticity correction of that measurement buffer at a given wavelength (mdeg), l represents the path length (cm), c represents the peptide concentration (mM), and n represents the number of amino residues (shown in Figure 8).

Discussion

Point 17: Change two to six: “In order to overcome these inherent shortcomings, in this study, two symmetric heptad repeat sequences were conjugated with a loop [...]” 

Response 17: We have revised the mistake (shown in Line 232).

Point 18: Same sentence. Change “conjugated” to “connected by”.

Response 18: We have revised the manuscript according to your comments and suggestions (shown in Line 232).

Point 19: Reconsider the statement: “Gram-negative bacteria are more sensitive to AMPs, because their cell walls are thinner and the outermost layer is LPS compared to Gram-positive bacteria [24]”. Gram-negative are not more sensitive, in fact there are more AMPs being specific for Gram-positive bacteria as seen in Malanovic N, Lohner K. Antimicrobial Peptides Targeting Gram-Positive Bacteria. Pharmaceuticals (Basel). 2016 

Response 19: We have changed the statement to “Compared to the single membrane and thick peptidoglycan layer of Gram-positive bacteria, double membrane structure of Gram-negative bacteria is a weaker barrier, so Gram-negative bacteria are more sensitive to AMPs.” (shown in Line 236-238)

    We wrote this statement based on the following references:

  “However, HV2 had strong antibacterial activity against Gram-negative bacteria and almost no bactericidal activity against Gram-positive bacteria. It has been shown that the double membrane structure of Gram-negative bacteria is a weaker barrier compared to the mono-membrane and thick peptidoglycan layers of Gram-positive bacteria; therefore, Gram-negative bacteria are more susceptible than Gram-positive bacteria” (Dong, N. Bioactivity and bactericidal mechanism of histidine-rich β-hairpin peptide against Gram-negative bacteria. Int. J. Mol. Sci. 2019, 20, 3954.)

   “Furthermore, extensive characterization showed that the double membrane of gram-negative bacteria is a less efficient barrier than the single membrane and the thick peptidoglycan layer of the gram-positive bacteria; consequently, gram-negative bacteria is more susceptible to permeabilization than gram-positive bacteria” ( Wang, J.J. High specific selectivity and membrane-active mechanism of the synthetic centrosymmetric α-helical peptides with gly-gly pairs. Sci. Rep. 2015, 5, 15963).

    We have provided the proper references on this statement in manuscript.

Point 20: Line 269: “promo ting”

Response 20: We have revised the mistake (shown in Line 260).

Point 21: The authors have used a more appropriate control for comparison of the results obtained with the designed peptides, but here in the discussion it would be useful to compare the results (MIC, hemolytic effect, stability) with a natural AMP as melittin. I would introduce some sentences comparing these values in order to know the relevance of the results obtained with LRpG.

Response 21: We have added these sentences: “In this study, we also compared a natural AMP (melittin) to LRpG in the GMMIC, MHC10, GMSI and stability results. Although exhibited a little better antibacterial activity than LRpG, melittin had strong hemolysis effect. By calculating the GMSI values (MHC10/GMMIC), we found that the GMSI of LRpG was 584 times higher than melittin. Furthermore, LRpG exerted a good conditional stability and additive effects with conventional antibiotics. Based on the results above, LRpG overcomes the main limiting factors of natural AMP (such as melittin) and has the potential to develop into a therapeutic agent.” in discussion (shown in 283-289).

Methods

Point 22: Salt sensitivity assay. To test the pH effect, the buffer solution was adjusted. What is the composition of this buffer? 

Response 22: We are very sorry for the incomplete description of this part. We have changed this part to “For pH sensitivity, peptides were treated with different pHs for 1 h. Briefly, peptides were diluted in deionized water, and the pHs of the solutions were adjusted to 4.0, 6.0, 8.0 and 10.0 with HCl or NaOH, respectively.” (shown in 380-382).

Reviewer 2 Report

The authors have addressed the issues that I had posed and, in my opinion, the manuscript could be accepted in its present form.

Author Response

Response to Reviewer 2 Comments

Comments and Suggestions for Authors:

The authors have addressed the issues that I had posed and, in my opinion, the manuscript could be accepted in its present form.

Response: We thank Reviewer 2 very much for favorable comments and constructive suggestions on our manuscript.

Reviewer 3 Report

This manuscript describes new AMPs , reports their antimicrobial action and some data of the mechanism of action. It is pertinent and, in principle seems interesting. Methods used are adequate and the results have interest. However,  the manuscript is poorly written and should be revised by someone with good level of English and with knowledge on how to write a paper. 

Some sentences are too long and it is hard to read any of them

A few examples:

L 101 The antimicrobial activity of designed peptides was summarized in Table 2: was---is L 101 and 102

. The results showed that the antimicrobial activity of the designed peptides against Gram-negative bacteria was better.

Remove . The results showed that

better than what? Gram positives?

If this is so

. The antimicrobial activity of the designed peptides was higher in Gram-negative than in Grampositive.

If the final sentence is this one, then some considerations should be made, since the authors wrote that the main advantage resides in the fact that AMPs do not need receptor and a significant higher activity in Gram negatives would be consistent with a binding in the OM. 

L 115 performed----determined

L 124 For Gram-negative bacteria, LRpG showed the highest GMTI (73.1) in all peptides tested, which means that LRpG showed greater selectivity for Gram-negative bacteria compared to human erythrocytes.

This sentence needs to be completely re-written, may not be undertood.

L 131-139

The whole paragraph is unnecessary.

The first sentence is material and methods “In order to examine the salt sensitivity, the antimicrobial activity of the designed peptides was 132 determined by levels of concentrations of different salts added subsequently.”

The second “The antimicrobial activity of designed peptides against Escherichia coli (E. coli) ATCC25922 was hardly effected by physiological concentrations of different salts except for Na+, which could decrease the antimicrobial 135 activity when increasing the MICs by 4-8 folds (Table 3) is purely a repetition of table 3 data.

The third and fourth “. The MICs of designed peptides against E. 136 coli ATCC25922 were not changeable, even peptides incubated at 100°C for 30 min, showing strong 137 thermal stability (Table 4). In addition, LRZZ and LRα could maintain the antibacterial activity at 138 different pH condition” is purely a repetition of table 4 data  and the final one is discussion.

L 139 “These results suggested that LRpG can maintain antimicrobial activity at different conditions” suggested----demonstrated

L 140 “Therefore, LRpG has good potential in clinical application.” Has----may have.

Data are not enough to ensure clinical application . Much more research would be required.

Table 3. What “without physiological salts” means? MH medium? If this is so it should be mentioned in the table

If I understood the method, the AMPs are treated with salts, temperature and  different pHs, but the MIC is determined in Standard conditions. This is unclear in the manuscript.

For example in table 4 legend:

“The MICs(µM) of the designed peptides against E. coli ATCC25922 in the presence 146 of the mentioned condition (thermal and pH)”

Did the authors determine MIC at 100 ºC????

Or at pH 10???

This is, obviously, not adequately written.

Synergy with Conventional Antibiotics

In my view there is no synergism since values are in allcases higher than 0,5 and in only one at the limit (0,5) It would be better to write

Additive effect of AMPs and Conventional Antibiotics

L 161.  “Most AMPs kill bacteria by destroying bacterial cell membrane, which is an important protective barrier for bacteria.” The cytoplasmic membrane is much more than a protective barrier. This may be used when referring to outer membrane but bacterial cell membrane is the site for almost 100 % of biochemical reactions and plays in bacteria the roles of almost all cell organelles of the eukaryotic cell.

Change the sentence by “Most AMPs kill bacteria by destroying bacterial membranes”

Author Response

Response to Reviewer 3 Comments

    We are truly grateful to your comments and thoughtful suggestions, those comments are valuable for revising and improving our paper. We have studied comments carefully and have made correction which we hope meet with approval. Revised portion are marked in red in the manuscript. The responds to the comments are as flowing:

Point 1: This manuscript describes new AMPs, reports their antimicrobial action and some data of the mechanism of action. It is pertinent and, in principle seems interesting. Methods used are adequate and the results have interest. However, the manuscript is poorly written and should be revised by someone with good level of English and with knowledge on how to write a paper. 

Response 1: We are very sorry for our incorrect writing and the manuscript has been revised by someone with good level of English and with knowledge on how to write a paper. We used “a red font” mode in the WORD to show the revise portion in the revised manuscript.

Point 2: L 101 The antimicrobial activity of designed peptides was summarized in Table 2: was---is 

Response 2: We have revised the mistake (shown in Line 101).

Point 3: L 101 and 102. The results showed that the antimicrobial activity of the designed peptides against Gram-negative bacteria was better. Remove. The results showed that.  better than what? Gram positives?  If this is so. The antimicrobial activity of the designed peptides was higher in Gram-negative than in Gram-positive. If the final sentence is this one, then some considerations should be made, since the authors wrote that the main advantage resides in the fact that AMPs do not need receptor and a significant higher activity in Gram negatives would be consistent with a binding in the OM. 

Response 3: We have changed the sentence to “The designed peptides showed better antimicrobial activity against Gram-negative bacteria than Gram-positive bacteria. (shown in Line 101-103), and explained this result in the discussion.  

    We have added the sentence “Compared to the single membrane and thick peptidoglycan layer of Gram-positive bacteria, double membrane structure of Gram-negative bacteria is a weaker barrier, so Gram-negative bacteria are more sensitive to AMPs.” in the discussion to explain this result (shown in Line 236-238).

    We added this sentence based on the following references:

   “However, HV2 had strong antibacterial activity against Gram-negative bacteria and almost no bactericidal activity against Gram-positive bacteria. It has been shown that the double membrane structure of Gram-negative bacteria is a weaker barrier compared to the mono-membrane and thick peptidoglycan layers of Gram-positive bacteria; therefore, Gram-negative bacteria are more susceptible than Gram-positive bacteria” (Dong, N. Bioactivity and bactericidal mechanism of histidine-rich β-hairpin peptide against Gram-negative bacteria. Int. J. Mol. Sci. 2019, 20, 3954.)

   “Furthermore, extensive characterization showed that the double membrane of gram-negative bacteria is a less efficient barrier than the single membrane and the thick peptidoglycan layer of the gram-positive bacteria; consequently, gram-negative bacteria is more susceptible to permeabilization than gram-positive bacteria” ( Wang, J.J. High specific selectivity and membrane-active mechanism of the synthetic centrosymmetric α-helical peptides with gly-gly pairs. Sci. Rep. 2015, 5, 15963).

    We have provided the proper references on this statement in manuscript.

Point 4: L 115 performed----determined.

Response 4: We have revised the mistake (shown in Line 115).

Point 5: L 124 For Gram-negative bacteria, LRpG showed the highest GMTI (73.1) in all peptides tested, which means that LRpG showed greater selectivity for Gram-negative bacteria compared to human erythrocytes.

This sentence needs to be completely re-written, may not be undertood.

Response 5: We have changed the sentence to “Among all tested peptides, LRpG showed the highest GMTI (73.1) against Gram-negative bacteria” (shown in Line 124-125).

Point 6: L 131-139 The whole paragraph is unnecessary.

The first sentence is material and methods “In order to examine the salt sensitivity, the antimicrobial activity of the designed peptides was determined by levels of concentrations of different salts added subsequently.”

The second “The antimicrobial activity of designed peptides against Escherichia coli (E. coli) ATCC25922 was hardly effected by physiological concentrations of different salts except for Na+, which could decrease the antimicrobial activity when increasing the MICs by 4-8 folds (Table 3) is purely a repetition of table 3 data.

The third and fourth “The MICs of designed peptides against E. coli ATCC25922 were not changeable, even peptides incubated at 100°C for 30 min, showing strong thermal stability (Table 4). In addition, LRZZ and LRα could maintain the antibacterial activity at different pH condition” is purely a repetition of table 4 data and the final one is discussion.

Response 6: We have removed the whole paragraph.

Point 7: L 139 “These results suggested that LRpG can maintain antimicrobial activity at different conditions” suggested----demonstrated

Response 7: We have revised the mistake (shown in Line 130).

Point 8: L 140 “Therefore, LRpG has good potential in clinical application.” Has----may have. Data are not enough to ensure clinical application. Much more research would be required.

Response 8: We have revised the mistake (shown in Line 131).

Point 9: Table 3. What “without physiological salts” means? MH medium? If this is so it should be mentioned in the table.

If I understood the method, the AMPs are treated with salts, temperature and different pHs, but the MIC is determined in Standard conditions. This is unclear in the manuscript.

For example, in table 4 legend:

“The MICs(µM) of the designed peptides against E. coli ATCC25922 in the presence of the mentioned condition (thermal and pH)” Did the authors determine MIC at 100ºC???? Or at pH 10??? This is, obviously, not adequately written.

Response 9: We are very sorry for the inadequately written of this part. We have changed “The control MICs were determined without physiological salts” to “The control MICs were determined in MHB medium without physiological salts” (shown in Line 134).

    We have changed “The MICs(µM) of the designed peptides against E. coli ATCC25922 in the presence of the mentioned condition (thermal and pH)” to “The MICs(µM) of the designed peptides against E. coli ATCC25922 after temperatures and pHs treatment” (shown in Line 138-139).

Point 10: Synergy with Conventional Antibiotics. In my view there is no synergism since values are in all cases higher than 0,5 and in only one at the limit (0.5). It would be better to write “Additive effect of AMPs and Conventional Antibiotics”.

Response 10: We have revised the manuscript according to your comments and suggestions (shown in Line 140).

Point 11: “Most AMPs kill bacteria by destroying bacterial cell membrane, which is an important protective barrier for bacteria.” The cytoplasmic membrane is much more than a protective barrier. This may be used when referring to outer membrane but bacterial cell membrane is the site for almost 100 % of biochemical reactions and plays in bacteria the roles of almost all cell organelles of the eukaryotic cell. Change the sentence by “Most AMPs kill bacteria by destroying bacterial membranes”

Response 11: We have revised the manuscript according to your comments and suggestions (shown in Line 153).

Round 2

Reviewer 3 Report

Authors have generated a revised version in which all comments to the original submission have been taken into account. I think the paper may be published.

Just a comment: 

At 8 μM, the peptides resulted in an outer membrane permeability of more than 75%. Moreover....

What means a permeability of more than 75%.

75% of what?

Or, is an increase of permeability?

Author Response

Response to Reviewer 3 Comments

We are truly grateful to your comments and thoughtful suggestions, these comments are valuable for revising and improving our paper. We have studied reviewer’s comments carefully and have made revision which marked in red in the paper. For English language and style, the manuscript was not only edited by MDPI, but also revised by someone with good level of English and with knowledge on how to write a paper. We tried our best to improve the manuscript, and hope that the correction will meet with approval. The responds to the comments are as flowing:

Point 1: At 8 μM, the peptides resulted in an outer membrane permeability of more than 75%. Moreover....    What means a permeability of more than 75%.  75% of what?  Or, is an increase of permeability?

Response 1: We are very sorry for our inaccurate description of the result. We have changed the sentence to “At peptides concentrations greater than 8 µM, the outer membrane permeability induced by LRpG and LRα was over 75%.” (shown in Line 160-161)

According to material method 4.8, the permeability of peptides to the Gram-negative bacterial outer membrane can be reflected by N-phenyl-1-naphthylamine (NPN) uptake [1-3]. We used 10 µg/mL polymyxin B as the positive control, the outer membrane permeability induced by LRpG and LRα of different concentrations was calculated by the following equation: 

NPN uptake (%) = (Fobs - F0)/ (F100 -F0) × 100%, where Fobs is the observed fluorescence of NPN with E. coli ATCC25922 cells at a given peptide concentration, F0 is the initial fluorescence of NPN with E. coli ATCC25922 cells, and F100 is the fluorescence of NPN with E. coli ATCC25922 cells in the addition of 10 µg/mL polymyxin B.

References

[1] Chou, S.L.; Shao, C.X.; Wang, J.J.; Shan, A.S.; Xu, L.; Dong, N.; Li, Z.Y. Short, multiple-stranded β-hairpin peptides have antimicrobial potency with high selectivity and salt resistance. Acta Biomater. 2016, 30, 78-93.

[2] Wang, J.J.; Chou, S.L.; Xu, L.; Zhu, X.; Dong, N.; Shan, A.S; Chen, Z. High specific selectivity and membrane-active mechanism of the synthetic centrosymmetric α-helical peptides with gly-gly pairs. Sci. Rep. 2015, 5, 15963.

[3] Dong, N.; Wang, C.S.; Zhang, T.T.; Zhang, L.; Xue, C.Y.; Feng, X.J.; Bi, C.P.; Shan, A.S. Bioactivity and bactericidal mechanism of histidine-rich β-hairpin peptide against Gram-negative bacteria. Int. J. Mol. Sci. 2019, 20, 3954.

This manuscript is a resubmission of an earlier submission. The following is a list of the peer review reports and author responses from that submission.

Round 1

Reviewer 1 Report

Comments to the authors

In the article “High cell selectivity and bactericidal mechanism of symmetric α-helical antimicrobial peptides centered on β-turn” the authors describe the design of six peptides and their physicochemical characteristics. The design of the peptides is inspired by the heptad repeats that conform the coiled coils. They also introduce a loop (called β-turn by the authors) with the aim of improving the cell selectivity of these peptides as previously shown for β-hairpin AMPs. In order to know more about the secondary structure of the peptides they performed circular dichroism experiments. As the aim of the work is to figure out if adding a β-turn (loop) can improve the cell selectivity of the peptides, the authors calculate the MIC values for gram negative and gram positive bacteria and measure the hemolytic activity against RAW264.7 and HEK293T cells. They also study the stability of the peptides in the presence of different salts and temperatures. The potential synergistic effect of the most promising peptide is tested and showed a 4-8 fold decrease in MIC for streptomycin and other antibiotics. In order to dig into the mechanism of action of the most promising peptide they performed permeability measurements, membrane potential measurements and DNA and LPS binding assays. The use of SEM in peptide-treated Escherichia coli cells allows the visualisation of the peptide effect on the surface of the bacteria. 

To be considered in the whole manuscript:

Extensive editing of English language and style is required.  Why do the authors use the term “antimicrobial peptides” in the title “Symmetric α-helical antimicrobial peptides centered on β-turn”. In the text no sign of having used an AMP sequence as a scaffold for the designing of the six peptides is shown. This could be observed e.g. in Ryan L et al . Anti-antimicrobial peptides: folding-mediated host defense antagonists. J Biol Chem. 2013. 288(28):20162-72.  The structural description of the designed peptides could be more accurate. In the text is written: “two symmetric α-helical coiled coils units and β-turn sequence”. In spite of designing heptad repetitions with hydrophobic residues in the a and d positions this cannot ensure that the structure is going to fold as a coiled-coil. If the aim of the design was to design peptides that fold as a coiled-coil it would be important to consider hydrophobic a/d pairs in conjunction with charged e/g pairs. See Ryan L et al (2013) “Anti-antimicrobial peptides” JBC 288: 20162-20172.  The structural description of the designed peptides could be more accurate. In the text is written: “two symmetric α-helical coiled coils units and β-turn sequence”. The β-turn consists of four amino acids with hydrogen bonds between the i and i + 3 residues. It cannot be said that the addition of two amino acids forms a β-turn. In this case it would be a loop. Why do the authors use melittin as a control? The similarities between melittin and the designed peptides are low. Melittin is longer (26 amino acids), less hydrophobic, it shows a poor cell selectivity, strong hemolytic activity against bacterial and mammalian cells and no evidence of coiled coil structure has been reported. The tetrameric structure of this peptide shows a toroidal channel with a central cavity diameter of 2.5-4.5 nm. Could the authors compare their results with another peptide closer to the designed peptides regarding the structure and/or sequence?

Introduction

Second paragraph: ”α-helical coiled coils and β-hairpin are common structures of AMPs”. Why do the authors claim that coiled coils are common structures in AMPs? The structural classification of AMPs includes a group of α-helical peptides but it is not reported that they fold as coiled coil. Second paragraph: “Some papers showed that symmetric amphiphilic β-hairpin AMPs centered on β-turn had a good cell selectivity“. The authors must rewrite this sentence. If the authors want to compare the designed peptides with antimicrobial peptides (as melittin) it would be useful to talk about the cationic and amphipatic nature of AMPs in the introduction. In the reference 12 Kumar, A et al (2016) the authors showed that an isoleucine residue at the d position of the heptad repeat of piscidin-1 contributed more prominently to the hemolytic and antiendotoxin properties of piscidin-1 than a valine residue at the same position, despite the similar hydrophobicities of the two amino acids. Have the authors considered the design of peptides by adding valine residues? Last paragraph (when the authors describe the rigid and flexible β-turns): reference 19 has to be changed to 20 (regarding flexible loops). Reference 25 must be added (regarding rigid loops). Last paragraph: If the introduction of arginines in the heptad sequence is done with the aim of increasing the positive charges, is there any explanation for adding arginines instead of lysines? Last paragraph: In the paper "Papanastasiou et al (2009) (number 21 in the references section) it is not shown any improvement of the antimicrobial activity after acetylation. Why do the authors claim that acetylation could add stability to the peptide structure? Could they add other references?

Results

2.1. Peptide design and characteristics

In Table1 the sentence “lower-case letter indicates the D-enantiomer of the proline” must be added. In Table 1 “(3): The level of perfectly amphipathic AMP is reflected by the relative hydrophobic moment of a peptide”. I would rewrite this sentence (e.g. “the relative hydrophobic moment of a peptide is its hydrophobic moment relative to that of a perfectly amphipatic peptide”).

2.2 Circular dichroism (CD) Spectroscopy

If the authors are investigating the selectivity of the designed peptides it would be more interesting to perform the CD in the presence of lipid mixtures that mimic bacterial (anionic unilamellar vesicles) and mammalian (zwitterionic unilamellar vesicles) membranes. This would provide interesting information about the folding of the designed peptides in these two environments.  Random coil is not a defined structural element but rather a conformation where the monomer subunits are oriented randomly while still being bonded to adjacent units. The typical signals show a minimum around 200nm. As the spectra don´t show this minimum the authors mustn ́t use this term. They could use “disordered conformations” instead.  “In membrane-mimetic environments […] with obvious negative peaks at 205nm and 220nm.” The peak around 222 nm is not evident. The authors must remove “obvious” and explain this phenomenon. Maybe the analysis between the θ​222/​θ208 ratio could help the authors in the explanation. As reported, single helices have a ratio of 0.9 while coiled coils have a ratio greater than 1.10 (Zheng, T. ​et al.​ Probing coiled-coil assembly by paramagnetic NMR spectroscopy. ​Org. Biomol. Chem.​ ​13​, 1159–1168 (2015)).  In the legend of Figure 1 is written “bule” instead of “blue”.

2.3 Antimicrobial activity

In Table 2, the authors calculate GM and TI mixing the Gram-negative and Gram-positive bacteria values. These global GM and TI values are not useful since the peptides exhibited very different values between the two types of bacteria. In Table 2: “A value of 256 μM was for TI calculation while…”. Add “was used” and change “while” by “when”.  As graph 2b shows the same behaviour in all the tested concentrations it would be useful to add the hemolytic effects (for one selected concentration) in table 2. In this way it would be easier to compare the cell selectivity of the peptides in bacteria and mammalian cells. 

2.4. Biocompatibility assays

Second line: “[...] and melittin was as a control”. Add “was used as a control”. Legend of Figure 2: “[...] based on t at least”. Remove “t”.

2.5. Condition Sensitivity assays

Table 3: Add the concentrations of the different salts used in the assay. Table 3: In the title “Physiological salts”, what does the number 2 indicate? Why do the authors use a different concentration for the salts employed in the measurements? (see material and methods). The authors must explain this fact. Have the authors tested different pH? As peptide structure could be dependent on pH this experiment could be very interesting.

2.6. Synergy with conventional antibiotics. 

Third line: “the fractional inhibitory concentration (FIC) index (FICI). Remove FIC or FICI.  FICI values range from 0.5 to 4. Which is the criteria followed by the authors in order to propose a value higher than 4? The FIC is calculated by comparing the value of the MIC of each agent alone with the combination-derived MIC. Antimicrobial combinations that result in a 4-fold reduction in the MIC compared with the MICs of agents alone are synergistic (FIC ≤ 0.5) (Doern CD. When does 2 plus 2 equal 5? A review of antimicrobial synergy testing. J Clin Microbiol. 2014 ;52:4124-8). With a decrease in MIC of more than 8 fold for ciprofloxacin and chloramphenicol, why is not considered as a synergistic effect?

2.7.1. Outer membrane permeability assay 

First sentence is too general. Could the authors explain the meaning of the sentence? Figure 3: What does a permeability greater than 100% mean? Showing the data for polymyxin B (control) in the graph would be useful.

2.7.2. Inner membrane permeability assay

First sentence: “The [...] was determined by measuring the cytoplasm β-galactosidase”. Add “activity” at the end of the sentence.  Last sentence: “This phenomenon was different from the results of outer membrane permeability”. Why are the authors claiming this?

2.7.4. Scanning electron microscopy (SEM) 

How can the authors assert that the pore formation is happening by visualisation of the SEM images?

2.7.5. DNA binding assay

The binding of a polycationic peptide to a polyanionic molecule as DNA is expected. What do want the authors show with this experiment?  If the gel doesn´t run longer (as MW marker indicates) is difficult to see the real effects of the experiment. Add the concentration of peptide for every lane (it is not indicated in “material and methods”). It would be needed to add a lane with the peptide alone in order to see if at high concentrations it remains stuck in the well of the gel or it is a real retardation effect.

2.7.6. LPS binding assay 

Second line: “ [...] (LAL) assay was to analyze […]”. Add “was used” or “was performed”. Regarding the ability of the peptide in binding LPS it would be useful to perform CD in the presence of different concentrations of LPS. Have the authors considered this option?

Discussion

Second line “[...] reduce bacterial drug resistance”. As the authors know, bacteria have developed some mechanisms to become resistant to antimicrobial peptides (see Maria-Neto S, de Almeida KC, Macedo ML, Franco OL. Understanding bacterial resistance to antimicrobial peptides: From the surface to deep inside. Biochim Biophys Acta. 2015 1848: 3078-88 (2015).” Third line: “[...] inherent defects”. Authors must remind the reader the problems in using peptides as therapeutic agents. Fourth line: “[...] a series of hybrid AMPs with two symmetric α-helical coiled coils unit and β-turn sequence element were designed [...]. Could the authors explain the use of the term “hybrid AMPs” in the context of the designed peptides? Is there any evidence that melittin forms a coiled coil? “In addition, the antibacterial activity of LRpG against E.coli ATCC25922 is decreased because of the obstruction of electrostatic interaction between [...]”. Change “obstruction” by “disruption”. “Thermal stability is very important for the clinical application of AMPs because many food and feed products need to be heated during processing”. Could the authors justify the introduction of this sentence in the discussion? “In addition to the membrane mechanisms mentioned above, LRpG can bind to intracellular material DNA at a concentration much higher than MIC concentration, suggesting that the DNA binding of LRpG may serve as a complementary bactericidal mechanism”. The EMSA assay performed does not provide enough information for claiming that DNA binding may serve as a complementary bactericidal mechanism.

Reviewer 2 Report

The manuscript describes peptide conjugates designed to exhibit antimicrobial activity and the authors have been able to identify a conjugate with an interesting antimicrobial activity profile.

Even though the manuscript, in my opinion, deserves publication, major issues should be addressed prior to publication. In particular:

1) The title is not clear. In my opinion, it is difficult to understand what the authors mean by "centered on b-turn"

2) Authors should explain the rationale behind the selection of D-Pro-Gly and Gly-Gly as b-turns.

3) Authors should change "hybridize" by "conjugate" throughout the manuscript. See for exemple line 75, the sentence "In this study, two symmetric α-helical coiled coils units and β-turn were hybridized." is not understandable. 

4) In line 88, "trifluoroethyl alcohol" should be replaced by trifluoroethanol.

5) In line 154, "nphenyl-1-naphthylamine" should be replaced by N-phenyl-1-naphthylamine.

6) In line 111, "rates" should be replaced by "percentatges".

7) Authors should include in the Supporting Information the HPLC, the ESI-MS and the HRMS results for all peptides.

The above issues are relevant but, in my opinion, the most important one is the English revision. All the manuscript should be extensively revised because there are many sentences that can not be understood. I have enclosed below just some examples:

Line 20. "In order to overcome these problems, 19 we designed a series of two symmetric α-helical coiled coils units and β-turn sequence element 20 hybrid AMPs based on the sequence template XXRXXRRzzRRXXRXX-NH2,"

Line 44. "However, more than 3000 AMPs have 44 been found that the development of AMPs as a therapeutic agent has seriously hindered because of 45 some limiting factors, such as manufacturing costs, cytotoxicity, poor stability and so on"

Line 52. "Previous studies mostly aimed at the influence of the different hydrophobic 52 amino acids on them in the "a" and "d" positions on antibacterial and cytotoxicity of AMPs, but there 53 have few studies on using this sequence element to design new AMPs"

Line 97. "The results showed 97 that designed peptides against Gram-negative bacteria was better in antimicrobial activity."

Line 129. "The antimicrobial activity of LRpG was decreased 129 by Na+ when increasing MICs by 4-fold."

Line 220. "The results may due to the cell walls of Gram-negative bacteria are thinner and the outermost layer is LPS with negatively charge compared to Gram- positive bacteria, that is why they are more sensitive to AMPs"

Line 223. "Previous studies demonstrated that proper positive charge was essential for antibacterial activity (+6~+7), which increased the positive charge of AMPs beyond the threshold, but antibacterial activity was no longer increased"

Reviewer 3 Report

The ms reports the synthesis of a new model of CAMPs presumably of interest. However,  the manuscript should be largely improved, shortened and copy-edited by some expert in scientific literature. Many sentences may not be understood.